# Topical Wound Treatment with a Nitric Oxide-Releasing PDE5 Inhibitor Formulation Enhances Blood Perfusion and Promotes Healing in Mice

**DOI:** 10.3390/pharmaceutics14112358

**Published:** 2022-10-31

**Authors:** Maya Ben-Yehuda Greenwald, Yu-Hang Liu, Weiye Li, Paul Hiebert, Maria Zubair, Hermann Tenor, Tobias Braun, Reto Naef, Daniel Razansky, Sabine Werner

**Affiliations:** 1Institute of Molecular Health Sciences, Department of Biology, ETH Zurich, Otto-Stern-Weg 7, 8093 Zurich, Switzerland; 2Institute for Biomedical Engineering and Institute of Pharmacology and Toxicology, Faculty of Medicine, University of Zurich, 8057 Zurich, Switzerland; 3Institute for Biomedical Engineering, Department of Information Technology and Electrical Engineering, ETH Zurich, 8092 Zurich, Switzerland; 4Topadur Pharma AG, Grabenstrasse 11A, 8952 Schlieren, Switzerland

**Keywords:** angiogenesis, nitric oxide, phosphodiesterase, optoacoustic microscopy, photoacoustic imaging, wound healing

## Abstract

Chronic, non-healing wounds constitute a major health problem, and the current therapeutic options are limited. Therefore, pharmaceuticals that can be locally applied to complicated wounds are urgently needed. Such treatments should directly target the underlying causes, which include diabetes mellitus, chronic local pressure and/or vascular insufficiency. A common consequence of these disorders is impaired wound angiogenesis. Here, we investigated the effect of topical application of a nitric oxide-releasing phosphodiesterase 5 inhibitor (TOP-N53)-containing liquid hydrogel on wound repair in mice. The drug-loaded hydrogel promoted re-epithelialization and angiogenesis in wounds of healthy and healing-impaired diabetic mice. Using a non-invasive label-free optoacoustic microscopy approach combined with automated vessel analysis, we show that the topical application of TOP-N53 formulation increases the microvascular network density and promotes the functionality of the newly formed blood vessels, resulting in enhanced blood perfusion of the wounds. These results demonstrate a remarkable healing-stimulating activity of topically applied TOP-N53 formulation, supporting its further development as a wound therapeutic.

## 1. Introduction

The skin fulfills various vital functions, including the regulation of body temperature, sensory reception, synthesis of vitamins and hormones, water balance, and protection against environmental insults. Therefore, its efficient repair after injury is of utmost importance. This is achieved by a dynamic and complex healing process [1,2,3]. Unfortunately, a large percentage of the population shows severe healing impairments, in particular, many aged individuals and patients with diabetes or those treated with immunosuppressive steroids [4,5]. These patients often develop ulcerative skin defects (chronic wounds), which mainly include venous and arterial ulcers, pressure ulcers and diabetic foot ulcers [4]. Such wounds are characterized by chronic inflammation and a failure to produce new tissue, whereby re-epithelialization and the formation of new blood vessels are particularly affected [4,6]. The treatment options for chronic wounds are still unsatisfactory and largely restricted to dressings and devices that help to provide a healing-promoting local milieu [7]. Importantly, very few clinical advances have been made in the past few decades in treating chronic, non-healing wounds [8], leaving an increased number of patients with few therapeutic options and causing a huge burden to the health care system [4,5,9]. The existing therapeutic approaches, including hyperbaric oxygen therapy, the usage of skin substitutes, negative pressure therapy and local delivery of platelet-derived growth factor, are limited in their efficacy, and currently, there is no approved small molecule compound for the treatment of problematic, non-healing wounds [10].

Previous studies revealed beneficial effects of nitric oxide (NO) donors and phosphodiesterase inhibitors on wound healing. For example, an NO-releasing polymer incorporated ointment promoted wound closure and several relevant wound healing parameters in mice [11] and application of the NO donor molsidomine reversed the impaired healing in diabetic rats [12]. The phosphodiesterase 5 (PDE5) inhibitor sildenafil (Viagra) promoted wound healing in healthy and experimental diabetic rats [13] and also improved the delayed wound healing in irradiated tissues [14]. To combine the positive effects of NO donors and PDE5 inhibitors, we previously developed TOP-N53, a small molecular weight NO-releasing phosphodiesterase 5 inhibitor (PDE5I). Remarkably, intradermal injection of this compound at the wound edge improved wound re-epithelialization and angiogenesis in healthy and healing-impaired diabetic mice without enhancing inflammation or scar formation [15]. Moreover, we showed the superiority of TOP-N53 in donating NO and inhibiting PDE5 compared with single-acting NO-donors or PDE5 inhibitors, resulting in enhanced levels of cGMP [15]. However, the translational potential of this compound remains a challenge, and it is as yet unclear if TOP-N53 indeed promotes the functionality of wound vessels, thereby allowing enhanced blood perfusion of the wound tissue. Furthermore, wound management in a clinical setting will require an efficient topical wound treatment. This would not be possible with TOP-N53 dissolved in dimethyl sulfoxide (DMSO) as used in our previously published study where TOP-N53 was delivered via intradermal injection [15]. Here, we produced and characterized an immediate release TOP-N53-containing topical wound formulation, using ingredients which are accepted for wound treatment by the regulatory authorities and compatible with wound standard of care. We demonstrate its ability to promote wound re-epithelialization and angiogenesis in healthy and healing-impaired diabetic (db/db) mice. Furthermore, the non-invasive and longitudinal monitoring of wound healing progression using a label-free optoacoustic microscopy (LSOM) approach in combination with automated vessel analysis shows that topical application of a TOP-N53 liquid hydrogel formulation promotes wound blood flow and enhances microvascular network density.

These results reveal that the TOP-N53 hydrogel formulation improves major deficiencies of poorly healing wounds and encourage further pre-clinical and clinical research with this compound.

## 2. Materials and Methods

### 2.1. Preparation and Characterization of a TOP-N53 Formulation in a Liquid Hydrogel

To manufacture 100 g of topical hydrogel formulation of TOP-N53 at a final concentration of 165 µM, an aqueous gel was prepared by adding hydroxyethyl cellulose (HEC; 0.25 g) (CAELO, Hilden, Germany) in portions to 12 mM phosphate buffer (pH 6.7) (Rades GmbH, Hamburg, Germany). Butylated hydroxytoluene (0.006 g) (Sigma-Aldrich, St. Louis, MO, USA) and benzyl alcohol (2.0 g) (PanReacAppliChem, Barcelona, Spain) were added to polyethylene glycol (PEG) 400 (70.0 g) (BASF, Ludwigshafen, Germany), heated to 50 °C and cooled down after dissolution. TOP-N53 (9.97 mg; 16.5 µmol) was dissolved in the PEG 400 mixture (heated to 50 °C for 15 min and stirred for 2 h at room temperature). The PEG 400 mixture was added to the aqueous gel (equilibrated to room temperature at phase merging) to a final weight of 100 g and stirred with a spatula until a homogenous gel was obtained. The procedure to manufacture the vehicle formulation was identical, but without TOP-N53. The vehicle formulation is also abbreviated ‘HEC/PEG’ in the remainder of the manuscript. To manufacture hydrogels containing 1, 3 or 10 µM TOP-N53, the HEC/PEG formulation with TOP-N53 at 165 µM prepared as described above was appropriately diluted in HEC/PEG. The hydrogel formulations at different TOP-N53 concentrations keep the compound in solution, and no particles are formed. Moreover, TOP-N53 was stable in the hydrogel for at least two weeks. Formulations were prepared freshly for all wound healing experiments.

The viscosity of TOP-N53 liquid hydrogel formulation was measured using a Modular Compact Rheometer MCR 102 (Anton Paar GmbH, Graz, Austria). The measurements were done using cone plate geometry (diameter 50 mm) with 5 min resting phase. Fifty measurements with varying duration of each measurement (logarithmic: 100 sec start value und 0.1 sec end value, segment duration: 759.862 sec) were performed. Rheometer data analysis software RheoCompass (Anton Paar GmbH, Graz, Austria) was used for analysis. Measurement of TOP-N53 release from the hydrogel formulation was performed using the two-chamber Rapid Equilibrium Dialysis (RED) plate (Thermo Scientific Scientific^TM,^ Waltham, MA, USA). TOP-N53 hydrogel formulation was transferred to the donor chamber in the RED plate, while the receiver chamber was filled with human blood plasma (ZHBSD, Rütistrasse 19, CH-8952 Schlieren, Switzerland; (https://www.zhbsd.ch/kunden/preislisteagb/, accessed on 3 April 2022). Agreement for the use of the plasma was obtained from the responsible Cantonal Ethics Committee (BASEC Number 2020-00513, issued by the Cantonal Ethics Committee Zurich, Switzerland). The plasma was diluted 1:1 with normocin (1:500, Invivogen, San Diego, CA, USA)-treated simulated interstitial fluid buffer (SISF, pH 7.4) composed of 117 mM NaCl, 3 mM KCl, 2.8 mM CaCl_2_ × 2H_2_0, 1 mM MgCl_2_, 27 mM NaHCO_3_, 1 mM K_2_HPO_4_ and 0.5 mM Na_2_SO_4_. The RED plate was then sealed with an aluminum foil, placed into a 37 °C incubator and incubated under continuous shaking at 200 rpm. After 24 h, the complete liquid of the receiver chamber was taken. The sample was then prepared for the subsequent liquid chromatography–mass spectrometry (LCMS) measurement using the solid phase extraction (SPE, protein precipitation Sirocco plate, Waters Corp, Milford, MA, USA) method (Mass spectrometry AB Sciex API 4000 LC/MS/MS Triple Quad, reverse mode, InfinityLab Poroshell 120 Bonus-RP, 2.1 × 50 mm, 2.7 µm, narrow bore LC column, solvent A: 2 mM ammonium formate in water + 0.1% formic acid, solvent B: 2 mM ammonium formate in methanol/acetonitrile (50/50; *v*/*v*) + 0.1% formic acid, flow rate: 400 µL, run time: 5.2 min). During the experiment, deuterated TOP-N53D4 was spiked into the receiver chamber and used as an internal standard, creating a calibration curve for further TOP-N53 content quantification.

### 2.2. Cell Culture

Human primary foreskin keratinocytes were cultured in CnT-Prime Epithelial Cell Culture Medium (CELLnTEC, Bern, Switzerland). Human foreskin fibroblasts were cultured in Dulbecco’s Modified Eagle’s Medium DMEM (Sigma-Aldrich, St. Louis, MO), supplemented with 10% fetal bovine serum (FBS) and penicillin/streptomycin.

### 2.3. MTT Assay

The MTT assay was used to evaluate the cytotoxicity of the different treatments. Cultures were incubated for 22–24 h with 0.1 or 1 µM TOP-N53 or vehicle dimethyl sulfoxide (DMSO) (0.01%), or with vehicle liquid hydrogel formulation (HEC/PEG) or TOP-N53 liquid hydrogel formulation at a concentration of 1 or 10 µM TOP-N53 in 1:10, 1:100, or 1:1000 dilutions as indicated. One or ten µM TOP-N53 liquid hydrogel formulation was diluted 1:10 in cell culture medium to a final concentration of 0.1 or 1 µM TOP-N53 in liquid hydrogel formulation, respectively. For the treatment of human primary keratinocytes with TOP-N53 in a 0.01% (final) DMSO formulation, TOP-N53 or vehicle were sequentially diluted in cell culture medium under sterile conditions from 10 mM solution of TOP-N53 in 100% DMSO to their respective final concentrations in the assay (0.1 or 1 µM TOP-N53 in 0.01% DMSO or 0.01% DMSO (vehicle)). Cultures were then incubated for 0.5 h with MTT (5 mg/mL in PBS) at 37 °C/5% CO_2_, and the absorbance at 590 nm was measured using a GloMax^®^Discover reader (Promega, Fitchburg, MA, USA).

### 2.4. Measurement of cGMP Levels in Human Platelets

Buffy coats from human peripheral blood were commercially acquired from the Zurich Red Cross Blood Donation Service (ZHBSD, Rütistrasse 19, CH-8952 Schlieren, Switzerland) (https://www.zhbsd.ch/kunden/preislisteagb/ accessed on 3 April 2022). Buffy coats were only released from ZHBSD once negative results from assays for HCV, HIV, Treponema pallidum, HBV, HEV, HAV, ParvoB19, and West Nile Virus were available. Agreement for the use of commercially acquired buffy coats from the competent Cantonal Ethics Committee was obtained (BASEC Number 2020-00513, issued by the Cantonal Ethics Committee Zurich, Switzerland). Experimental work was done in a Biosafety Level-2 (BSL2) environment. Work with human buffy coats was filed as a project with the Swiss Federal Office for the Environment (FOEN, BAFU) at https://www.ecogen.admin.ch/public/, application A203119-00. The protocol for isolation of washed platelets was adopted from Gambaryan et al., with modifications [16]. Buffy coats (50 mL) were diluted 3.2-fold with Buffer A composed of 34.8 mM trisodium citrate, 150 mM NaCl (pH 8.4) and centrifuged at 200× *g* for 10 min. The supernatant comprising platelet-rich plasma (PRP) was removed and ACD solution (85 mM trisodium citrate, 71 mM citric acid, 111 mM D-glucose (pH 4.4)) was added at one tenth of the final volume followed by addition of 0.01 U/mL (final concentration) apyrase. After centrifugation at 200× *g* for 5 min to separate remaining white and red blood cells, the PRP was removed, placed into new tubes, and centrifuged at 700× g for 15 min. The diluted platelet-poor plasma (PPP) was aspirated, and the platelet pellet washed once in buffer D (120 mM NaCl, 30 mM D-glucose, 12.9 mM trisodium citrate (pH 6.5)) at 700× *g* for 15 min. The buffer supernatant was removed by aspiration, and platelets were resuspended in modified Tyrode (134 mM NaCl, 2.9 mM KCl, 12 mM NaHCO_3_, 0.36 mM NaH_2_PO_4_, 5 mM 4-(2-hydroxyethyl)-1-piperazineethanesulfonic acid (HEPES), and 5 mM D-glucose. Modified Tyrode in the current study was Tyrode supplemented to a final concentration of 0.2% (*w*/*v*) human serum albumin, 0.5 mM MgCl_2_, 0.01 U/mL apyrase. Platelets were counted in a modified Neubauer chamber and adjusted to 1.8 × 10^9^ cells/mL. Washed platelets (final concentration 4.5 × 10^8^ cells/mL) were pre-incubated for 10 min with the soluble guanylate cyclase stimulator riociguat (1µM, #HY-14779, MCE, Lucerna Chem AG, Lucerne, Switzerland) and the selective PDE2 inhibitor BAY 60-7550 (100 nM, HY#14-992, MCE, Lucerna Chem AG; IC50 rhPDE2A at 71 pM) and then incubated together with TOP-N53 in different formulations (see below). Following a 2 h incubation time, 20 µL 2N HCl were added. Supernatants from a 5 min 1000× *g* centrifugation step were frozen at −80 °C. Total platelet cGMP (pmol/mL) content was determined using a commercially available ELISA (Direct cGMP ELISA, ADI-901-014, Enzo Life Sciences ELS, Lausen BL, Switzerland), following the instructions of the manufacturer.

Test Items and formulations were as follows:

*TOP-N53, DMSO formulation.* TOP-N53 was diluted from a 10 mM stock solution in 100% DMSO to a final concentration of 3 µM in 0.1% DMSO/modified Tyrode in the final assay well.

*TOP-N53, Hydrogel formulation*. TOP-N53 was diluted from a 165 µM stock solution in HEC/PEG formulation (0.25% hydroxyethylcellulose, 70% PEG400 as main components, see Table 1 for a complete description of the ingredients) in modified Tyrode to an intermediate concentration of 30 µM and next 10-fold in the assay well to a final concentration of 3 µM TOP-N53. This final concentration was chosen based on the in vivo results, which identified an optimum at 3 µM (see Results, Section 3). Final concentrations of PEG400 and hydroxyethylcellulose in the assay well were 1.27% and 0.0045%.

*Vehicle hydrogel formulation*. The hydrogel vehicle was diluted 5.5-fold and finally 10-fold in the assay well.

In the experiments, DMSO was adjusted to a final concentration of 0.3% in all assay wells.

### 2.5. Animals and Wound Healing Experiments

Mouse maintenance and animal experiments had been approved by the local veterinary authorities (Kantonales Veterinäramt Zurich, Switzerland). Full-thickness excisional wounds (5 mm diameter) were generated on the back skin of female C57BL/6JRj, BKS(D)-Leprdb/JorlR, or SKH-1 mice at the age of 9–11 weeks. One day prior to wounding, the back skin was shaved and a layer of depilatory cream (Veet, Reckitt Benckiser, Heidelberg, Germany) was applied. After 2–5 min, the cream was cleared away, and the skin was cleaned with water and 70% ethanol. Mice were anesthetized in an induction chamber using 2–4% isoflurane in an oxygen/air mix for imaging experiments or by intraperitoneal injection of ketamine/xylazine (100 mg ketamine/5–10 mg xylazine per kg body weight) for wound healing experiments. Immediately and three days post-wounding, 20 µL of TOP-N53 liquid hydrogel formulation (final TOP-N53 formulation concentrations were 1, 3, 10 or 165 µM) or vehicle liquid hydrogel formulation were applied topically on the wound. As an example, the 3 µM TOP-N53 formulation corresponds to 0.000181%, and 36.28 ng were applied per wound. All four wounds of one individual mouse received an identical treatment. At the perimeter of the wound, 10 µL of Mastisol (Ferndale Laboratories, Fernadale, MI, USA) were applied to improve the adherence of the Tegaderm dressing (Tegaderm, 3M Deutschland GmbH, Neuss, Germany)), which was used to cover the wounds. Mice were single-housed during the experiments. They were sacrificed at day 5 post-wounding, and the wound tissue was excised and analyzed as described below.

### 2.6. Histology

Wounds were excised and fixed overnight with either 4% paraformaldehyde (PFA) or 95% ethanol/1% acetic acid, followed by tissue processing and paraffin embedding, or they were directly frozen in tissue freezing medium (Leica Microsystems, Heerbrugg, Switzerland). Sections of 7 μm thickness from the middle of the wounds were stained with hematoxylin and eosin (H&E) or with Herovici stain [17] or used for further immunohistochemistry/immunofluorescence analysis. Stained sections were photographed using a Zeiss AxioImager.M1 microscope equipped with Zen Pro software (Zeiss, blue edition, 3.2) (Carl Zeiss AG, Oberkochen, Germany) to control the Axiocam MRm camera or using the Pannoramic 250 Slide Scanner (3D Histech, Budapest, Hungary). Analysis of the various wound parameters [15] and staining quantifications were performed using ImagePro^®^ Plus software (Media Cybernetics Inc., Rockville, MD, USA).

### 2.7. Immunohistochemistry and Immunofluorescence Staining

Paraffin sections were dewaxed and rehydrated using a xylene/ethanol gradient. PFA-fixed sections were then incubated in citrate buffer (pH 6.0) at 95 °C for 1 h for antigen retrieval. Frozen sections were fixed with cold acetone. Sections were blocked with PBS containing 12% BSA for 1 h at room temperature, followed by incubation with primary antibodies. For immunohistochemical staining, a biotin-conjugated secondary antibody was used, and bound antibodies were detected using the Vectastain ABC kit and the diaminobenzidine peroxidase substrate kit (both from Vector Laboratories, Burlingame, CA, USA). For immunofluorescence analysis, slides were incubated at room temperature for 1 h with Cy3-conjugated secondary antibodies (Jackson ImmunoResearch Laboratories, Inc., West Grove, PA, USA) and counterstained with Hoechst 33342 (Sigma-Aldrich, St. Louis, MO). The antibodies used, their dilutions and the incubation conditions are listed in Table 2.

### 2.8. Large-Scale Optoacoustic Microscopy (LSOM) 

A previously described label-free large-scale optoacoustic microscopy (LSOM) imaging technique [18] was used to monitor changes in microvascular networks in mouse dorsal skin during the wound healing process. Briefly, LSOM visualizes the intrinsic optical absorption contrasts of living mammalian tissues primarily resulting from hemoglobin (Hb). A pulsed nanosecond laser (Onda, Bright Solutions, Italy) with 532 nm wavelength was used to generate optoacoustic responses from tissues. The induced broadband ultrasound (US) signals of vessels were recorded by a customized spherically focused polyvinylidene fluoride (PVDF) ultrasound sensor. The acquired signals were then processed to render three-dimensional images of microvasculature and further analyzed to extract vascular metrics using custom-developed algorithms in Matlab 2020b [18,19].

### 2.9. Large-Scale Dorsal Skin Imaging

A dorsal imaging mount (DIM) [18] was used to conduct non-invasive dorsal skin imaging. Mice were anaesthetized, placed on the DIM, and their body temperature was maintained at 37 °C by a built-in self-regulated heating pad. Eye protection cream was applied and left on the eyes during the entire imaging procedure. The coarse-resolution LSOM images covering both dorsal skin wounds over 30 × 11 mm² lateral field of view (FOV) were acquired with 20 µm pixel size, while a 5 µm pixel size was employed for the fine-resolution images for individual wounds over small areas (7 × 7 mm), showing the detailed microvascular features. The maximum laser per-pulse energy was 900 nJ at a maximum repetition rate of 12 kHz [18,19].

### 2.10. Automatic Vessel Segmentation and Analysis Algorithm (AVSA) for Skin Vasculature

An automatic vessel segmentation and analysis algorithm (AVSA) [18] was used to quantify the metrics of skin microvascular networks for the LSOM datasets. Hb content was calculated directly based on the sum of raw LSOM image intensity values divided by the total imaged area (the wound region was excluded), aiming to eliminate any potential bias introduced by the contrast enhancement and filtering procedures. Fill fraction was estimated using the total blood vessel area divided by the total imaged area for evaluating the skin vasculature distribution surrounding the wounds [19]. To precisely analyze other vascular parameters such as vessel diameter, LSOM volumetric datasets were first divided into two volumes, i.e., superficial and deeper vessels based on their relative depth from the skin surface, by means of a customized segmentation algorithm [18]. These two volumes were projected into 2D maximum intensity projections (MIPs) and analyzed individually using the AVSA, and the vascular parameters were then combined for statistical assessment. Otherwise, the overlapping areas/branches of small-superficial and large-deeper vessels would introduce potential false identification of skin vasculature [18]. To further identify each blood vessel and its corresponding vascular parameters, LSOM datasets of individual wounds (i.e., the fine-resolution images) were contrast enhanced, denoised and smoothed, and then underwent image thresholding, binarization and vessel skeletonization procedures by using the AVSA. Number of identified vessels, vessel diameter, vessel length and vessel tortuosity (i.e., sum of all angles divided by vessel length) were all calculated for assessment. Vessel parameters per wound were calculated and presented with the mean values ± standard error for the LSOM image statistics. A histogram of vessel diameters was also presented for day 5 post-wounding, including all identified vessels in each treatment group. The AVSA also provides the position information of identified vessels for each individual wound. Thus, we generated the vessel-parameter heatmaps by integrating specific parameters with their corresponding vessel position information, presenting the changes surrounding the wounds over time. To generate the heatmaps, vessels were grouped into 15 × 15 pixels over 7 × 7 mm² field of view (FOV) based on their position information. Every pixel then represents the total vessel count and average set of vessel parameters, for example, diameter calculated for all wounds.

### 2.11. Statistical Analysis

Statistical analyses were performed using GraphPad Prism 9/9.3.1 software (GraphPad Software Inc., La Jolla, CA, USA). Mann–Whitney U test was used for comparison of two groups. For analysis of wound closure, a two-sided Fisher’s exact test was applied. To study the dose response of TOP-N53 liquid hydrogel formulation, ordinary one-way ANOVA, Dunnet multiple comparisons test or One sample *t* test (mentioned in the text) was applied. To evaluate the effect of TOP-N53 liquid hydrogel formulation on wound blood perfusion, a two-way ANOVA, Bonferroni’s multiple comparisons test or unpaired two-tailed t-test (mentioned in the text) were used. Quantitative results are expressed as mean ± standard error of the mean (SEM). * *p* ≤ 0.05, ** *p* ≤ 0.01, *** *p* ≤ 0.001; **** *p* ≤ 0.0001.

## 3. Results

### 3.1. Preparation of a Non-Toxic TOP-N53 Liquid Hydrogel Formulation, Which Does Not Impair the Wound Healing Process

TOP-N53 is a unique dual-acting NO donor and PDE5I, which was specifically designed for local wound treatment [20]. The small molecule compound (molecular weight 604.68 g/mol) is highly lipophilic as reflected by a calculated logarithm of the partition coefficient (clogP) of 3.79 and is practically insoluble in water. Therefore, it is suitable for topical delivery to normal skin, but requires a carrier formulation to enable wound treatment. Hydrogel formulation was chosen for this purpose due to its abilities to support wound hydration, aid in oxygen penetration, and create an optimal wound healing environment that promotes cell proliferation and migration [21]. It also keeps TOP-N53 in solution at pharmacologically effective doses used in this study, and therefore, it can diffuse through the wound tissue. We prepared a liquid hydrogel formulation with ingredients approved by the regulatory authorities for topical administration in humans (Table 1). It is compatible with the chemical features of TOP-N53 [15] and allows its incorporation up to a concentration of 165 µM with relatively low viscosity (0.8684 [Pa*s]). Using an in vitro assay, we found that the hydrogel allows the efficient release of TOP-N53 in human blood plasma diluted 1:1 with simulated interstitial fluid buffer (Figure 1A). Importantly, TOP-N53 was active following its release as reflected by the increased total cGMP levels.

TOP-N53 liquid hydrogel formulation at 10-fold dilution had only a minor effect on the viability of primary human keratinocytes following 22–24 h of incubation (Figure 1C). A higher dilution of the formulation was required for the treatment of primary human foreskin fibroblasts because of mild toxicity for this cell type (Figure 1C, right panel).

To test if the formulation affects the wound healing process, vehicle liquid hydrogel formulation or saline solution were dripped topically on full-thickness excisional wounds of C57BL/6 mice, followed by the application of skin adhesive at the wound perimeter and attachment of occlusive wound dressing. This treatment was performed immediately as well as at day 3 after wounding (Figure 2A,B). Histomorphometric analysis of wound sections showed similar outcomes of healing and wound appearance with saline or vehicle liquid hydrogel formulation treatment (Figure 2C–I), emphasizing the safety of the formulation under the experimental conditions. Herovici staining, which stains young and mature collagen fibers in blue and purple, respectively [17], did not reveal obvious differences in the collagen-positive area in the granulation tissue between the vehicle formulation and saline treatment groups. Similar areas of young and mature collagen fibers and their ratio were observed, further emphasizing that the formulation does not affect the healing process (Figure 2C (right) and Figure 2J–M).

### 3.2. Topical TOP-N53 Formulation Increases Keratinocyte Proliferation and Wound Angiogenesis in Healthy Mice

To first test the effect of the TOP-N53 liquid hydrogel formulation on the healing of full-thickness excisional wounds in healthy mice, we chose day 5 after wounding for our analysis because of our previous experience with intradermal injection of TOP-N53 [15] and because this time point represents the peak of wound granulation tissue formation and re-epithelialization in the excisional wound model that we used [3].

TOP-N53 at 1 and 10 µM was applied in the formulation using the same procedure and treatment regimen as for the vehicle (Figure 3A,B). The treated mice did not show weight loss throughout the experiment (Appendix A). A slightly higher percentage of the TOP-N53 formulation-treated wounds were closed at day 5 compared with the vehicle formulation-treated wounds, although the difference was not statistically significant (Figure 3A,B). As expected for healthy mice in which the wound healing process is highly efficient, TOP-N53 treatment had no significant effect on re-epithelialization, length and thickness of the wound epidermis, wound contraction and granulation tissue area (Figure 3C–G). This was confirmed by semi-quantitative wound scoring (Appendix A). However, TOP-N53 formulation-treated-wounds showed increased proliferation of keratinocytes in the wound epidermis as demonstrated by Ki67 immunostaining, and the difference was significant at 1 µM TOP-N53 (Figure 3H–I). A mild, but non-significant increase in cell proliferation was also observed in the granulation tissue below the epithelial tongue and in the wound bed (middle of the wound) (Figure 3H,J,K). Treatment with TOP-N53 formulation caused only a very mild increase in CD68^+^ macrophages and Ly6G^+^ neutrophils in the granulation tissue, and only at the highest concentration (Appendix A).

The granulation tissue area, which stained positive for Meca32, a vascular endothelial cell marker, was significantly larger in the wounds treated with 10 µM TOP-N53. This was observed in the entire granulation tissue and in particularly at the wound edges, where angiogenesis is initiated (Figure 3L–O). Moreover, vessel maturation was increased at the wound edges as reflected by the co-staining of Meca32 and the vascular smooth muscle cell marker α-smooth muscle actin (Figure 3L,P–R). No obvious difference in the number of myofibroblasts was detected in the different treatment groups at the wound edges and in the entire granulation tissue as reflected by a similar area covered by α-smooth muscle actin-positive cells outside the vessels. However, myofibroblasts were more abundant in the wound bed of TOP-N53 formulation-treated vs. vehicle-treated-wounds (Appendix A).

### 3.3. Topical TOP-N53 Formulation Promotes Wound Re-Epithelialization and Angiogenesis in Healing-Impaired Diabetic Mice

Next, we tested the therapeutic efficacy of TOP-N53 liquid hydrogel formulation in mice with genetically determined type II diabetes (*db/db* mice), an established mouse model for impaired wound healing [22]. The diabetic condition of these mice was confirmed by analysis of blood glucose levels (Appendix A). We then generated full-thickness excisional wounds in these mice and treated them topically with vehicle formulation or TOP-N53 formulation at different concentrations according to the treatment procedure and regimen shown in Figure 2A,B. Neither TOP-N53 formulation at different concentrations nor vehicle formulation treatment affected the body weight during the treatment period (Appendix A). At day 5, most wounds were not fully re-epithelialized (Figure 4A,B), with the exception of two wounds treated with 3 µM TOP-N53, which were closed (Figure 4B). Histomorphometric analysis of the H&E-stained 5-day wound sections revealed that the impaired re-epithelialization that is characteristic for wounds in *db/db* mice (compare Figure 3C and Figure 4C) was significantly improved upon treatment with 3 µM TOP-N53 (Figure 4C). TOP-N53 3 µM formulation seemed to be an optimal treatment under these experimental conditions, as the wound re-epithelialization rate showed a “bell”-shaped-like curve and was no longer increased at higher concentrations. Wound re-epithelialization is achieved by a combination of keratinocyte migration and proliferation. A very mild increase in keratinocyte migration in the TOP-N53 treated wounds was suggested based on the length of the wound epidermis (Figure 4D). The thickness of the latter was only affected following treatment with a higher concentration of TOP-N53 formulation (Figure 4D,E). Wound contraction was mildly, but non-significantly increased upon treatment of wounds with TOP-N53 3 µM formulation (Figure 4F), while the area of granulation tissue was not altered (Figure 4G). However, TOP-N53 had a strong effect on the vasculature. There were significantly more vascular endothelial cells at the wound edges and in the entire granulation tissue in wounds treated with TOP-N53 3 µM vs. vehicle formulation, as reflected by Meca32 staining (Figure 4H–J), demonstrating that TOP-N53 3 µM formulation enhances wound angiogenesis.

Taken together, these results show that topically applied TOP-N53 in liquid hydrogel formulation is most efficient in diabetic mice when applied at a concentration of 3 µM, which promoted wound re-epithelialization and angiogenesis in these animals.

### 3.4. Topical Application of TOP-N53 Liquid Hydrogel Formulation Enhances Wound Blood Flow and Microvascular Network Density in SKH-1 Mice

Our previous and new data showed that TOP-N53 enhances wound re-epithelialization and angiogenesis in both healthy and healing-impaired diabetic mice upon intradermal injection [15] or topical application in hydrogel formulation (this study). In addition, it directly promoted endothelial cell migration and tube formation in vitro [15]. However, it was so far unclear if wound vascular blood flow is indeed improved. To address this question, we performed label-free non-invasive longitudinal imaging of the wound microvascular network using LSOM in SKH-1 hairless mice. These mice are particularly suitable for the visualization of wound angiogenesis, because they lack hair and melanin, which can interfere with imaging, but show normal wound healing [23]. SHK-1 mice were treated with vehicle formulation or with the optimized concentration of TOP-N53 formulation (3 µM) using the treatment regimen of the previous experiments (Figure 2B). Consistently, the mouse body weight was not affected during the experiment (Appendix A). The volumetric LSOM images were recorded at different time points for both treatment groups and all mice (Appendix A). Following image acquisition, skin or wound vasculature were analyzed using an automatic vessel segmentation and analysis algorithm (AVSA) to allow quantification of the relevant metrics [18,19]. Since LSOM is based on light absorption by hemoglobin (Hb), we first quantified the Hb content. This parameter is based solely on the sum of raw image intensity values, divided by the total imaged area, avoiding any image post-processing effects. We found significantly more intense signals and higher Hb content in the compound formulation-treated wounds at day 5 post-injury compared with the vehicle formulation-treated wounds (Figure 5A,B). Similarly, the fill fraction, which directly evaluates the skin vasculature distribution (e.g., calculated as the total blood vessel area divided by the total imaged area), was increased at the same time point (Figure 5C). The latter metric was even highly significant (** *p* = 0.008) when calculated using an unpaired two-tailed t-test. It is important to note that fill fraction reflects blood volume, which was shown to be correlated with blood flow [24,25]. A further detailed vascular analysis was performed following segmentation of the volumetric imaging data into two groups, according to the vessel size and their depth in the skin/wound. This was performed to increase the accuracy of our automatic vessel analysis by minimizing any 3D overlapping areas/branches of the vessels that interfere with other vessels, which may cause false identification of normal skin vasculature. For convenience purposes, we highlighted the surface vessels in orange and the deep vessels in blue, while the wound regions were blocked out (Figure 5D). To emphasize the formation of functional vessels over time, we created heat maps for functional vessel number and vessel diameter from a representative wound of each group at multiple time points (Figure 5E). There was a clear difference in both vessel number and vessel diameter between the treatment groups at day 5 post-injury (Figure 5E–G). At this time point, an obvious increase in vessel number was seen in the TOP-N53-treated wounds (Figure 5F). These vessels were smaller in diameter, likely indicating active vessel sprouting (Figure 5G). The reduction in vessel diameter at day 5 post-wounding in the TOP-N53-treated wounds was even more pronounced when considering the vessel diameter measured in intact skin. The full distribution of the individual vessel diameter at day 5 post-injury is shown in Figure 5H, revealing the significant difference between the treatment groups and showing high numbers of smaller vessels for the TOP-N53-treated wounds. In contrast, there was no difference in vessel tortuosity (e.g., sum of all angles of each vessel divided by its vessel length) between the treatment groups (Figure 5I). Note that, although the peaks of diameters of newly formed vessels in TOP-N53 treated wounds ranged from 16–22 µm at day 5 post-injury in the distribution curve (Figure 5H), the mean vessel diameter, which is calculated based on each wound (Figure 5G), reflects the mean value of individual vessel diameters, including the big vessels with a diameter of up to 100 µm (Figure 5H). This causes a shift towards a slightly higher mean value. 

Complementary histomorphometric analysis of the imaged wounds at day 5 after wounding showed a mild, but non-significant increase in wound re-epithelialization in the TOP-N53 group. There were no obvious changes in wound epidermis length or thickness, wound contraction, or granulation tissue area (Appendix A). Consistent with the results obtained in other mouse strains and with the imaging data, the Meca32-positive areas at the wound edges, in the wound bed and in the entire granulation tissue were significantly larger in the TOP-N53-treated mice (Appendix A). These results confirm the potent effect of TOP-N53 formulation on wound angiogenesis and show the excellent correlation between histomorphometric and LSOM data.

## 4. Discussion

Chronic, non-healing wounds have a significant economic impact on healthcare systems and are associated with high patient morbidity and mortality [4,5,9]. In such wounds, a variety of factors prevent healing, resulting in persistent inflammation and deficiencies in keratinocyte migration, fibroblast proliferation and migration, matrix deposition, and vascularization [4]. The toolset of treatment options available to clinicians is limited in its efficacy and often requires complex treatment procedures [7,8,26]. Therefore, new therapeutic approaches for the treatment of chronic wounds are required to improve the healing process and the patients’ quality of life.

Studies from our laboratory have shown that TOP-N53, which combines NO-driven stimulation of cGMP synthesis with reduced cGMP degradation through inhibition of PDE5, respectively, promotes different healing parameters in healthy and healing-impaired diabetic mice [15]. This was enabled through the compound’s action on the major skin resident cells, thereby accelerating wound re-epithelialization, granulation tissue formation and angiogenesis without inducing excessive inflammation and scar formation [15]. Ideally, this compound should be applied topically to poorly healing wounds [8].

Here, we show that a topically applied liquid hydrogel formulation allowing efficient release of TOP-N53 promotes wound re-epithelialization and angiogenesis. Remarkably, this was associated with an increase in wound microvascular network density and consequently, enhanced blood flow in the wound granulation tissue. By contrast, the effect of the TOP-N53 hydrogel on the immune cell response was only very mild, which is consistent with the findings obtained with intradermally injected TOP-N53 [15].

An important effect of TOP-N53 formulation was the promotion of wound re-epithelialization, which was seen upon intradermal injection of TOP-N53 [15] or following its local wound application in the HEC/PEG hydrogel formulation (this study). During wound re-epithelialization, keratinocytes undergo a partial epithelial-mesenchymal transition (EMT), which is important for the repair process [27]. In the future, it will be of interest to determine if EMT is affected by TOP-N53 treatment. The similar effects achieved with injected or topically applied TOP-N53 on re-epithelialization demonstrate the suitability of the liquid hydrogel formulation (HEC/PEG) for topical application, which is of key relevance for future therapeutic application. This was confirmed when we analyzed its effect on wound angiogenesis. In healthy mice, keratinocyte proliferation was most efficiently promoted at a concentration of 1 µM (20 µL hydrogel), while the optimal concentration for wound angiogenesis was 10 µM (20 µL hydrogel). This is consistent with results from others showing NO-mediated promotion of keratinocyte proliferation at an optimal NO-donor concentration [28] and may indicate a higher sensitivity of wound keratinocytes to the compound formulation treatment as compared to wound blood vessel cells.

The most striking effect exerted by TOP-N53 liquid hydrogel formulation was the enhanced blood flow combined with an increased wound microvascular network density. This is particularly important, as diabetic ulcers are characterized by low vascularity and capillary density [29,30], often as a consequence of compromised NO production and the resulting reduction in cGMP levels [4,31,32]. This is also consistent with our previous study showing a direct effect of TOP-N53 on endothelial cell migration and tube formation [15]. Here, we show that this effect is functionally relevant in vivo. By taking advantage of non-invasive longitudinal imaging of the wound microvascular network using LSOM, we demonstrate the ability of TOP-N53 in the HEC/PEG hydrogel formulation to promote functional vessel formation and to enhance wound blood perfusion. This is of paramount importance, as these vessels support the wound with nutrients and oxygen and also allow the invasion of immune cells and removal of hazardous agents, thereby promoting the repair process. Thanks to the non-invasive nature of LSOM, we were able to non-invasively and longitudinally monitor the therapeutic effects of TOP-N53, while the wound region of interest could still be collected after the last imaging time point for histological evaluation. The observation of an increased number of smaller, but functional blood vessels 5 days post-wounding in TOP-N53 formulation-treated wounds fits with the normal wound healing kinetics, where new blood vessels sprout and grow, and robust angiogenesis starts approximately 2–3 days post-wounding and peaks at day 5–7, followed by vessel regression through apoptosis [33]. Our work also demonstrates the suitability of the LSOM approach to study the effect of different compounds on blood vessel formation. This opens new avenues for the use of this new technology, which was recently used for the analysis of the normal wound healing process [18] and the effect of a *VEGFA* transgene on the vasculature in normal and wounded skin [19].

To this end, various key players in the wound healing process have been identified, including growth factors and cytokines [4,34,35]. Some of them have been used in translational studies, e.g., with the aim to enhance wound vascularization [33]. Our data suggest the use of a synthetic, stable, small molecular weight compound, TOP-N53, which may rescue the endothelial dysfunction apparent in chronic wounds and promote the formation of new functional vessels in skin wounds. When formulated in a liquid hydrogel, this compound can be used for topical application on wounds. Hydrogel formulations as presented in this study will be particularly suitable for wounds that allow a daily treatment, such as digital ulcers that result from poor blood flow due to vasoconstriction in systemic sclerosis patients [36]. Therefore, the results presented here encourage further translational research, including the development of delivery systems that allow application of TOP-N53 in a time-controlled manner and appropriate dose for improved treatment of chronic, non-healing skin wounds.

## Figures and Tables

**Figure 1 pharmaceutics-14-02358-f001:**
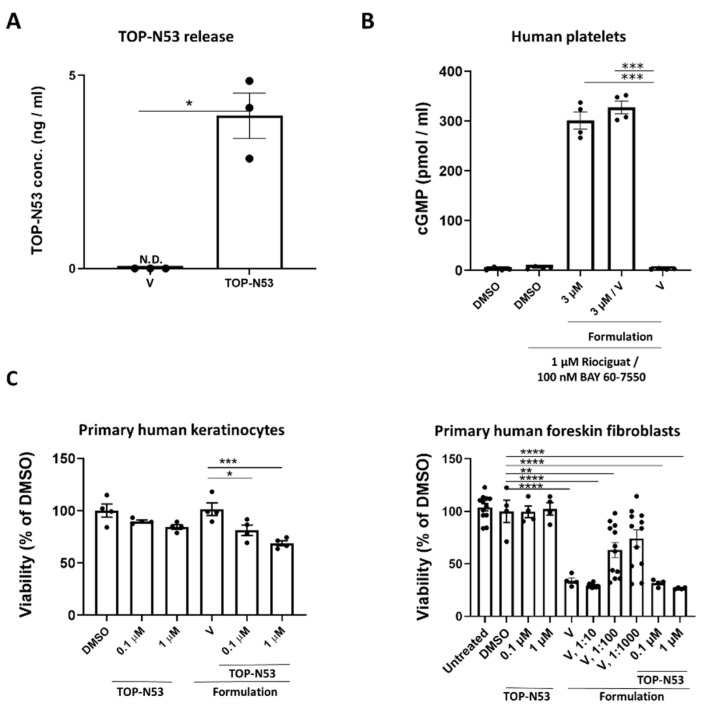
A non-toxic liquid hydrogel formulation allows release of active TOP-N53. (**A**) TOP-N53 concentration as measured by LCMS multiple reaction monitoring in human blood plasma diluted with simulated interstitial fluid buffer (1:1) following 24 h incubation in a rapid equilibrium dialysis device system. N = 3 per treatment group. N.D.: not detectable; V: Vehicle. (**B**) Washed human platelets were incubated with riociguat (1 µM) and BAY 60-7550 (100 nM) for 10 min and then treated with 3 µM of TOP-N53 hydrogel formulation or vehicle hydrogel formulation (V) or vehicle hydrogel formulation and 3 µM of TOP-N53 dissolved in DMSO for 2 h. Incubation was terminated by adding 20 µL of 2 N HCl. Total cGMP (pmol/mL) was measured by ELISA. N = 4 independent experiments using platelets from different donors. Primary human keratinocytes ((**C**), left) or foreskin fibroblasts ((**C**), right) were incubated for 22–24 h with 0.1 or 1 μM TOP-N53 in DMSO (0.01%) or 0.01% DMSO only, or with vehicle liquid hydrogel formulation (V) or TOP-N53 at final concentrations of 0.1 or 1 µM in liquid hydrogel formulation or with 10-, 100- or 1000-fold dilutions of the liquid hydrogel formulation in culture medium as indicated and analyzed for cell viability using MTT assay. N = 4–12 technical replicates per treatment group. Bars indicate mean +/− SEM. * *p* ≤ 0.05, ** *p* ≤ 0.01, *** *p* ≤ 0.001, **** *p* ≤ 0.0001 (Ordinary one-way ANOVA, one sample *t* test for data shown in Figure 1A).

**Figure 2 pharmaceutics-14-02358-f002:**
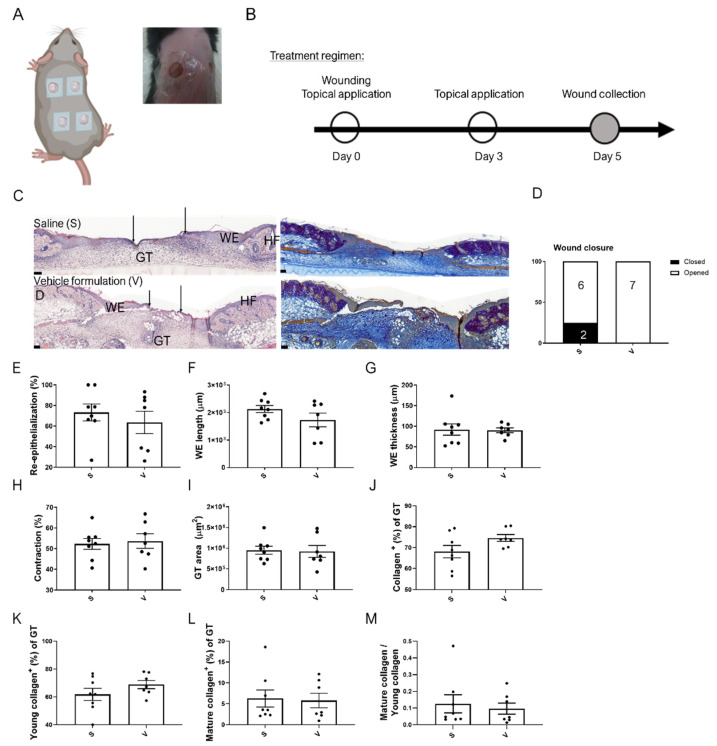
The liquid hydrogel formulation has no adverse effects on wound healing. (**A**) Scheme illustrating the topical wound treatment (left) and a representative topically treated wound covered with Tegaderm (right). The scheme was created using BioRender.com. (**B**) Treatment regimen for wound healing experiments. (**C**) Representative photomicrographs of hematoxylin/eosin (left)-and Herovici (right)-stained paraffin sections from 5-day wounds of healthy mice treated with saline or vehicle liquid hydrogel formulation. Magnification bars: 100 μm. D: Dermis, Es: Eschar, GT: Granulation tissue, HF: Hair follicle, WE: wound epidermis. Black arrows indicate the tips of the epithelial tongues. (**D**) Percentages of open (white bars) and closed (black bars) 5-day wounds based on histological sections. n = 8 (saline; S) or n = 7 (vehicle; V) wounds. (**E**) Percentage of wound re-epithelialization, including open and closed wounds. n = 7–8 wounds. Length (**F**) and average thickness (epidermis area/epidermis length) (**G**) of the wound epidermis (WE). n = 7–8 wounds. (**H**) Percentage of wound contraction based on the initial wound length (5 mm). n = 7–8 wounds. (**I**) Area of granulation tissue (GT). n = 7–8 wounds per treatment group. Percentage of collagen-positive GT area (**J**), percentage of young (blue) (**K**) or mature (purple) (**L**) collagen-positive GT area and ratio of mature (purple)-to-young collagen (blue) positive GT area (**M**) based on Herovici staining. n = 7–8 wounds. Bars indicate mean +/− SEM.

**Figure 3 pharmaceutics-14-02358-f003:**
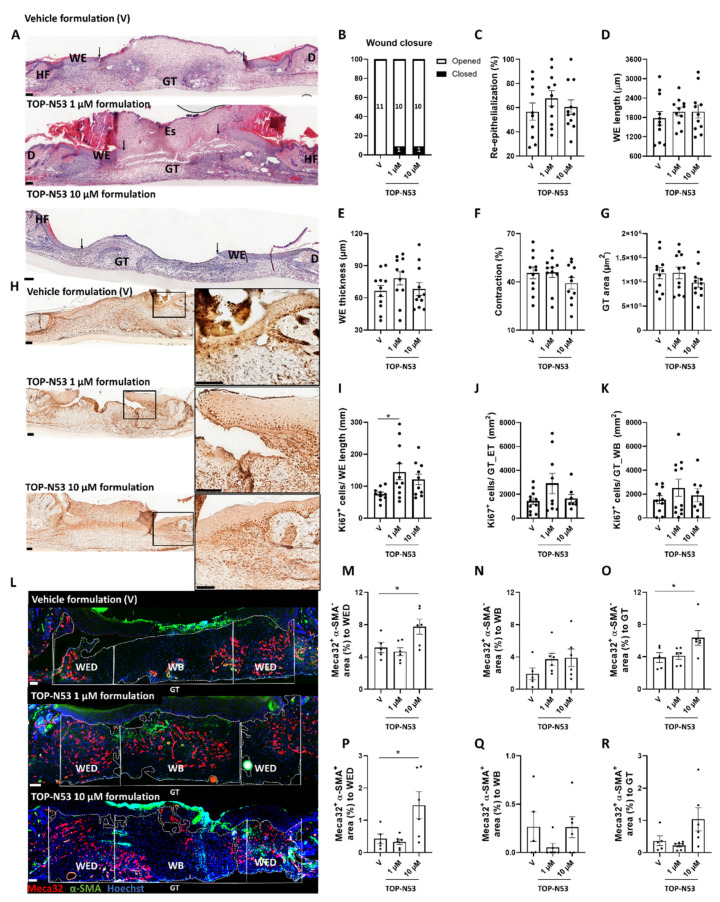
TOP-N53 liquid hydrogel formulation promotes keratinocyte proliferation and angiogenesis in wounds of healthy C57BL/6J mice. (**A**) Representative photomicrographs of hematoxylin/eosin-stained paraffin sections from 5-day wounds of mice treated with vehicle hydrogel formulation (V) or TOP-N53 1 or 10 µM liquid hydrogel formulation. Magnification bars: 100 μm. D: Dermis, Es: Eschar, GT: Granulation tissue, HF: Hair follicle, WE: wound epidermis. Black arrows indicate the position of the epithelial tongue. (**B**) Percentages of open (white bars) and closed (black bars) 5-day wounds based on histological sections. n = 11 (vehicle; V); n = 11 (1 µM) or n = 11 (10 µM). (**C**) Percentage of wound re-epithelialization, including open and closed wounds, n = 10–11 wounds per treatment group. (**D**,**E**) Length (**D**) and average thickness (epidermis area/epidermis length) (E) of the wound epidermis (WE). n = 11 wounds. (**F**) Percentage of wound contraction based on the initial wound length (5 mm). n = 11 wounds. (**G**) Area of GT. n = 11 wounds. (**H**) Representative photomicrographs of wound sections stained for Ki67. Magnification bars: 100 μm. Ki67^+^ cells per mm wound epidermis (**I**) or mm^2^ granulation tissue area below the epithelial tongue (**J**) or in the wound bed (**K**). n = 10–11 wounds. (**L**) Representative photomicrographs of wound sections stained for Meca-32 (red) and α-smooth muscle actin (SMA) (green) and counterstained with Hoechst (blue). White dashed lines define the different areas of the granulation tissue. Magnification bars: 100 μm. Percentage of GT tissue area at the wound edge area (WED) stained positive for Meca32 (**M**) or in the wound bed (WB) (**N**) or total GT (**O**). n = 5–6 wounds. Percentage of GT area at the wound edge (WED) stained positive for Meca32 and α-SMA (**P**) or in the wound bed (WB) (**Q**) or total GT (**R**). n = 5–6 wounds. All bars represent mean +/− SEM. * *p* ≤ 0.05; ordinary one-way ANOVA (all graphs, except analysis of open vs. closed wounds for which Fisher’s exact test was used).

**Figure 4 pharmaceutics-14-02358-f004:**
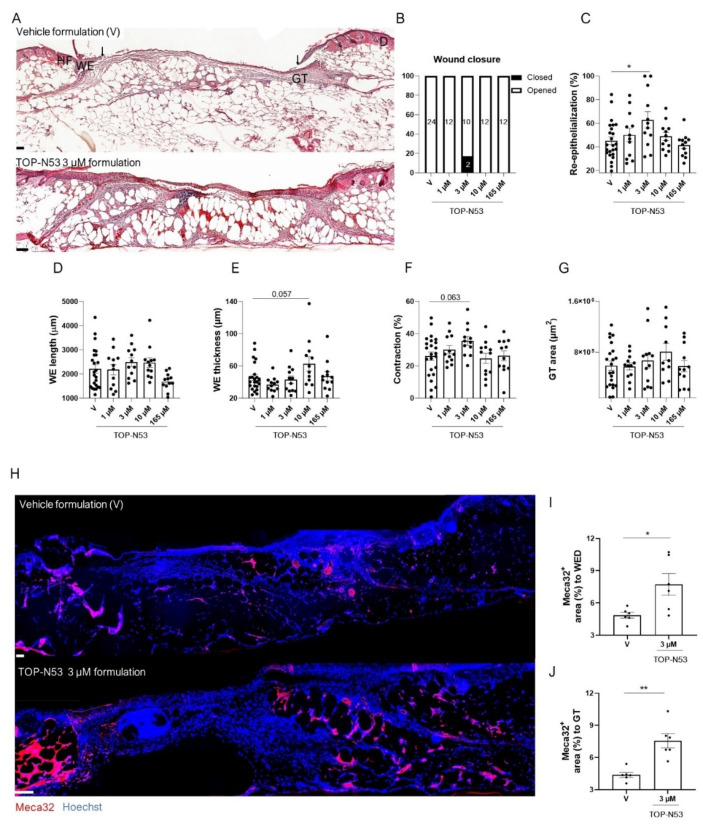
TOP-N53 liquid hydrogel formulation promotes wound re-epithelialization and angiogenesis in healing-impaired diabetic mice. (**A**) Representative photomicrographs of hematoxylin/eosin-stained paraffin sections from 5-day wounds of diabetic *db/db* mice treated with vehicle hydrogel formulation (V) or TOP-N53 3 µM liquid hydrogel formulation. Magnification bars: 100 µm. D: Dermis, GT: granulation tissue, HF: Hair follicle, WE: wound epidermis. (**B**) Percentages of open (white bars) and closed (black bars) 5-day wounds based on histological sections. n = 24 (vehicle; V); n = 12 (1 µM), n = 12 (3 µM), n = 12 (10 µM) and n = 12 (165 µM) wounds. (**C**) Percentage of wound re-epithelialization, including open and closed wounds. n = 12–24 wounds per treatment group. Length (**D**) and average thickness (epidermis area/epidermis length) (**E**) of the wound epidermis (WE). n = 12–24 wounds. (**F**) Percentage of wound contraction based on the initial wound length (5 mm). n = 12–24 wounds. (**G**) Area of GT. n = 12–24 wounds. (**H**) Representative photomicrographs of wound sections stained for Meca-32 (red) and counterstained with Hoechst (blue). Magnification bar: 100 μm. Percentage of Meca32-positive GT area at the wound edge (WED) (**I**) or in the total GT (**J**). n = 5–6 mice per treatment group. All bars represent mean +/− SEM. * *p* ≤ 0.05, ** *p* ≤ 0.01; ordinary one-way ANOVA (all graphs, except analysis of open vs. closed wounds for which Fisher’s exact test was used, and angiogenesis analysis where 2 groups were compared and Mann–Whitney U test was used).

**Figure 5 pharmaceutics-14-02358-f005:**
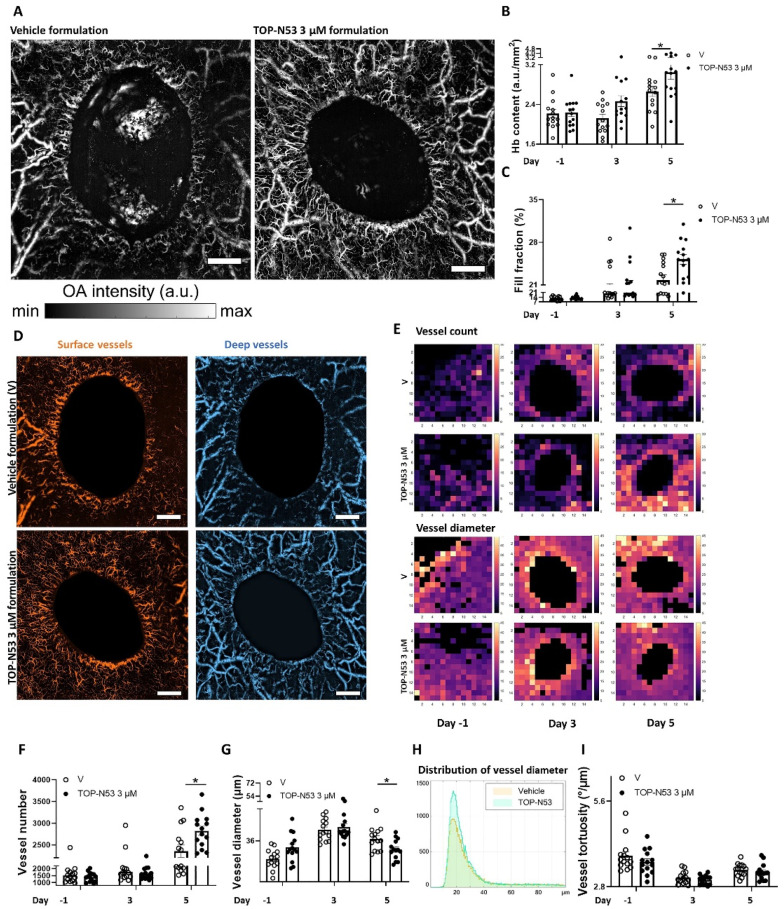
TOP-N53 liquid hydrogel formulation promotes wound blood flow and increases the number of small wound vessels in SKH-1 mice. (**A**) Representative LSOM images of 5-day wounds from SKH-1 mice treated topically either with vehicle hydrogel formulation (V) or TOP-N53 3 µM liquid hydrogel formulation. Magnification bars: 1 mm. The colormap represents optoacoustic signal intensity in arbitrary units (a.u.). (**B**,**C**) Analysis of hemoglobin (Hb) content (a.u./mm^2^) (**B**) or percentage of fill fraction (**C**) based on LSOM images acquired pre- and post-wounding at different time points. n = 14 wounds per treatment group. V: vehicle. (**D**) Representative LSOM images of wound vessels segmented into superficial (orange) and deeper (blue) vessels based on their size and relative depth to the skin surface. Magnification bars: 1 mm. (**E**) Spatial-averaged heatmaps (each pixel corresponds to 0.46 × 0.46 mm^2^ area) showing a qualitative assessment of vessel number and diameter per pixel pre- and post-wounding at different time points. n = 14 wounds. Analysis of vascular parameters including number of vessels (**F**), vessel diameter (**G**), distribution of vessel diameter (**H**) and vessel tortuosity (**I**). All parameters were calculated based on LSOM images acquired pre- and post-wounding at different time points. n = 14 wounds. All bars represent mean +/− SEM. Statistics is shown for day 5. * *p* ≤ 0.05 (Two-way ANOVA and Bonferroni multiple comparisons test).

**Table 1 pharmaceutics-14-02358-t001:** Composition of liquid hydrogel formulations: Vehicle and TOP-N53, 165 µM.

	Vehicle	165 µM TOP-N53
Ingredient	(%) (*w*/*w*)	(%) (*w*/*w*)
Polyethylene glycol 400 (PEG 400)	70	70
Benzyl alcohol	2	2
Butylated hydroxytoluene	0.006	0.006
Hydroxyethyl cellulose (HEC)	0.25	0.25
Phosphate buffer 12 mM pH 6.7	27.744	27.734
TOP-N53	0	0.01

PEG 400: Polyethylene glycol 400; HEC: Hydroxyethyl cellulose.

**Table 2 pharmaceutics-14-02358-t002:** Antibodies used for immunohistochemistry or immunofluorescence staining.

Antibody	Source	Dilution	Incubation Conditions	Identifier
Rabbit anti-Ki67	Abcam, Cambridge, UK	1:200	15 min at RT	Cat#Ab15580; RRID: AB_443209
Biotinylated anti-rabbit IgG	Jackson ImmunoResearch	1:1000	30 min at RT	Cat#111-065-003; RRID: AB_2337959
Rat anti-Meca32	BD Biosciences, Franklin Lakes, NJ	1:1000	Overnight at 4 ℃	Cat#553849; RRID: AB_395086
Mouse anti-α-smooth muscle actin-FITC	Sigma-Aldrich	1:500	Overnight at 4 ℃	Cat#F3777; RRID: AB_476977
Anti-rat-Cy3	Jackson ImmunoResearch	1:200	30 min at RT	Cat#715-165-150; RRID: AB_2340666

## Data Availability

Not applicable.

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
