# Peer review of "Topical Wound Treatment with a Nitric Oxide-Releasing PDE5 Inhibitor Formulation Enhances Blood Perfusion and Promotes Healing in Mice"

_pharmaceutics, 2022, doi:10.3390/pharmaceutics14112358_

Round 1

Reviewer 1 Report

This manuscript by Greenwald et al. examined the effect of topically applied nitric oxide-releasing PDE5 inhibitor (TOP-N53)-containing liquid hydrogel on wound healing in mice and found that TOP-N53-hydrogel formulation promoted keratinocyte proliferation, re-epithelialization and angiogenesis in mouse skin wounds. In addition, they found improved microvasculature network and functionality of newly formed blood vessels promoting the wound healing in healthy mice as well as diabetic mice. Overall, the manuscript is very well written. The authors have presented very interesting and novel studies supporting the concept that TOP-N53 hydrogel formulation would improve the chronic wound healing process by maintaining the skin structural integrity and functions. The authors are encouraged to address the following which would significantly improve the visibility and the comprehensiveness of this body of work.

1)   Did the author evaluate TOP-N53 release kinetics in mice in-vivo? It would be interesting if the authors could quantify "TOP-N53 input" from formulation to the viable skin that would enable a key component of dermal pharmacokinetics to be characterized.

2)   TOP-N53 formulation treatment appear to cause an increase in CD68+ macrophages and Ly6G+ neutrophils in the granulation tissue area. How do authors interpret inflammatory activity of this treatment? It would be interesting if authors could examine the abundance of MMPs, iNOS, IL-10, TGF-beta, TNF-alpha, IL-1 beta, and CXCLs mRNA transcripts. This would help to understand the microenvironment of wound after TOP-N53 formulation treatment.

3)   Epithelial-mesenchymal transition (EMT) is critical in wound healing process. It would be interesting if the authors could discuss as well as examine the effects of TOP-N53 formulation on EMT in wounded skin by analyzing EMT-associated molecules. 

Author Response

Reviewer 1:

Comments and Suggestions for Authors

This manuscript by Greenwald et al. examined the effect of topically applied nitric oxide-releasing PDE5 inhibitor (TOP-N53)-containing liquid hydrogel on wound healing in mice and found that TOP-N53-hydrogel formulation promoted keratinocyte proliferation, re-epithelialization and angiogenesis in mouse skin wounds. In addition, they found improved microvasculature network and functionality of newly formed blood vessels promoting the wound healing in healthy mice as well as diabetic mice. Overall, the manuscript is very well written. The authors have presented very interesting and novel studies supporting the concept that TOP-N53 hydrogel formulation would improve the chronic wound healing process by maintaining the skin structural integrity and functions. The authors are encouraged to address the following which would significantly improve the visibility and the comprehensiveness of this body of work.

1)  Did the author evaluate TOP-N53 release kinetics in mice in-vivo? It would be interesting if the authors could quantify "TOP-N53 input" from formulation to the viable skin that would enable a key component of dermal pharmacokinetics to be characterized.

Our response: In vivo release kinetics has not been done. We would like to point out that TOP-N53 at 3 µM is in solution in the hydrogel formulation and there are no particles. We now clarify this issue in the text (Materials and Methods, 2.1).

2) TOP-N53 formulation treatment appear to cause an increase in CD68+ macrophages and Ly6G+ neutrophils in the granulation tissue area. How do authors interpret inflammatory activity of this treatment? It would be interesting if authors could examine the abundance of MMPs, iNOS, IL-10, TGF-beta, TNF-alpha, IL-1 beta, and CXCLs mRNA transcripts. This would help to understand the microenvironment of wound after TOP-N53 formulation treatment.

Our reply: The increase in the number of CD68+ macrophages was not significant at any concentration and the increase in neutrophils was also very mild and only significant at the highest concentration. Therefore, there is clearly no major effect of TOP-N53 on wound inflammation. We have further clarified this point in the text, and we also mention that this is consistent with our previous data, which showed that TOP-N53 did not affect wound inflammation after intradermal injection at the wound edge. We agree that a more detailed molecular analysis of the wound tissue would be of interest. Unfortunately, we do not have wound tissue or RNA available for this purpose and we would have to do a new wound healing experiment. This is not possible within the 10-day time frame that was given for the revision. In addition, a major change in the levels of these cytokines is unlikely, because this would cause a much stronger effect on wound immune cells.

3) Epithelial-mesenchymal transition (EMT) is critical in wound healing process. It would be interesting if the authors could discuss as well as examine the effects of TOP-N53 formulation on EMT in wounded skin by analyzing EMT-associated molecules.

Our reply: Keratinocytes at the edge of skin wounds only undergo a partial EMT. We agree that this is important for wound healing, and therefore, this is an interesting aspect for future studies. We now mention this in the text (page 18, lines: 621-627) and we cite a review on (partial) EMT in wound healing (Haensel and Dai, 2018).

Reviewer 2 Report

Comments to the Author

The article describes the wound treatment with a nitric oxide-releasing PDE5 in-2 hibitor formulation enhances blood perfusion and promotes 3 healing in mice. However, there are some issues need to be fixed.

1.      The content of the Introduction section is less. It should introduce the related research background, especially the work of others.

2.      It is suggested to add a preparation diagram of the material in this paper, so as to quickly understand the preparation process.

3.      What is the stability of TOP-N53 in liquid hydrogel?

4.      Not clear the concentration of liquid hydrogel for the preparation of TOP-N53, Hydrogel formulation.

5.       The incorporation of TOP-N53 in polymer matrix should be determined through AFM or SEM.

Author Response

Reviewer 2:

The article describes the wound treatment with a nitric oxide-releasing PDE5 in-2 hibitor formulation enhances blood perfusion and promotes 3 healing in mice. However, there are some issues need to be fixed.

  1. The content of the Introduction section is less. It should introduce the related research background, especially the work of others.

       Our reply: We have extended the introduction and included information on the effect of NO donors and PDE5 inhibitors on wound healing.

  1. It is suggested to add a preparation diagram of the material in this paper, so as to quickly understand the preparation process.

       Our reply: The description of the preparation is described in detail in section 2.1, and therefore, we believe that a diagram wound not provide additional information.

  1. What is the stability of TOP-N53 in liquid hydrogel?

       Our reply: We now mention in Materials and Methods (2.1) that TOP-N53 in hydrogel is stable for at least 2 weeks. In addition, we mention that the hydrogel was always freshly prepared for the wound healing experiments.

  1. 4. Not clear the concentration of liquid hydrogel for the preparation of TOP-N53, Hydrogel formulation.

       Our reply: The concentrations of all components of the liquid hydrogel are shown in Table 1 of the revised manuscript.  

  1. The incorporation of TOP-N53 in polymer matrix should be determined through AFM or SEM.

       Our reply: As indicated in the manuscript, TOP-N53 at the used concentrations in the hydrogel is in solution (TOP-N53 was dissolved in PEG400 as indicated in 2.1). There are no particles in this formulation. Therefore, AFM or SEM has not been done.

Reviewer 3 Report

The manuscript entitled “Topical wound treatment with a nitric oxide-releasing PDE5 inhibitor formulation enhances blood perfusion and promotes healing in mice”, is the continuation of a previous work performed by the same group of authors. In this update, authors have demonstrated that a topical formula based on TOP-N53, instead of the injectable preparation, has the same ability to stimulate re-epithelialization and angiogenesis by a mechanism dependent of nitric oxide increase, as well as phosphodiesterase 5 inhibition. The active ingredient, per se, has been evaluated before and is well known because those biological activities, however the novelty of the present manuscript seems to be related to the hydrogel formula, molecule stability and activity in the formula. Besides, findings obtained in vivo by label-free optoacoustic microscopy correlated with histomorphological blood vessel evaluation. After reading the manuscript, this reviewer considers that the work has been performed in a flawless way. Data have been very well analyzed, and the conclusions have been supported by the obtained results.

However, there are several considerations that should be addressed to allow the readers understand and visualize the information appropriately.

1.      Please correct Dulbecco´s Modified Eagles Medium, where the apostrophe is missing (Eagle’s).

2.      This reviewer considers that the paragraph with the TOP-N53 concentration preparation (lines 185-191) is difficult to understand. Because the proposed serial dilutions between DMSO and vehicle come from different stock solutions, and it is not clear the final concentration that would be evaluated; sometimes in µM, others in %. Finally, these preparations do not match with the TOP-N53 concentration used in some experiments, such as in viability assays. I suggest authors declare the stock solutions concentrations and the final ones; always in the same units (µM or %).

3.      Why Table 2 is mentioned before Table 1? Please rearrange Tables 1 and 2, in order they appear as methods indicate.

4.      Please indicate the meaning of the acronym FOV (field of vision).

5.      Please indicate acronyms after every figure legend or at the end of the whole figure legend text. For example, Figure 1A has N.D. (I guess not detected) and "v", I suppose "vehicle", etc.

6.      Why TOP-N53 activity was evaluated at concentrations different to those assessed for viability? What is the cGMP expression by platelets when treated with 0.1 and 1 µM? (Figure 1B).

7.      Figure 1D. Why this chart does not include TOP-N53 treatment? The corresponding figure legend indicates that both, Figure 1C and 1D represent cell treatments with the different vehicles and formulations.

8.      Since riociguat is the active substance, it does not be written with a capital letter.

9.      Figure 1 legend. This reviewer considers that "(C,D)" is unnecessary, because the figures "C" and "D" are indicated immediately after. Same for the other figure legends.

10.   In some figures, the charts did not include asterisks depicting statistical significance. If the compared groups do not exhibit any significant difference please avoid the text “*P≤ 0.05, **P ≤ 0.01, ***P ≤ 0.001, ****P ≤ 0.0001” in the figure legend.

11.   This reviewer considers that Supplementary figures S1 C and D are relevant for the findings, I suggest to incorporate to Figure 3 or rearrange Figure 3 and create another figure where charts be grouped.

12.   Figure 3. Re-epithelialization does not seem be lower in the TOP-N53 10 µM formulation than the 1µM one. If you compare epidermal tongues (arrows) in the representative photomicrographs the difference is very clear.

13.   Figure 3. To be more explicit, please include dashed lines and acronym indicators (WED, WB and WED) in TOP-N53 1 and 10 µM treated tissues.

14.   Figure 5. Asterisks are too small to recognize them in the chart.

15.   This reviewer considers that Discussion section should include a brief analysis and comparison between DMSO and hydrogel formulations for TOP-N53.

16.   Figure S2. Please indicate surface and deep vessels above the LSOM images.

Author Response

Reviewer 3

  1. Please correct Dulbecco´s Modified Eagles Medium, where the apostrophe is missing (Eagle’s).

       Our reply: This has been corrected.

  1. This reviewer considers that the paragraph with the TOP-N53 concentration preparation (lines 185-191) is difficult to understand. Because the proposed serial dilutions between DMSO and vehicle come from different stock solutions, and it is not clear the final concentration that would be evaluated; sometimes in µM, others in %. Finally, these preparations do not match with the TOP-N53 concentration used in some experiments, such as in viability assays. I suggest authors declare the stock solutions concentrations and the final ones; always in the same units (µM or %).

       Our reply: We have further clarified this issue in the text, and we mention that the stock concentration of TOP-N53 in the hydrogel was 165 µM and the final concentration of TOP-N53 in the cGMP assay was 3 µM. This concentration was selected for the platelet experiments due to its optimal effects in the in vivo investigations. We have clarified this issue in the text. Concentrations of PEG400, hydroxyethylcellulose etc, DMSO are provided in percent, as usual.

  1. Why Table 2 is mentioned before Table 1? Please rearrange Tables 1 and 2, in order they appear as methods indicate.

       Our reply: This has been corrected.

  1. Please indicate the meaning of the acronym FOV (field of vision).

       Our reply: This information (field of view) has been included.

  1. Please indicate acronyms after every figure legend or at the end of the whole figure legend text. For example, Figure 1A has N.D. (I guess not detected) and "v", I suppose "vehicle", etc.

       Our reply: This has been done.

  1. Why TOP-N53 activity was evaluated at concentrations different to those assessed for viability? What is the cGMP expression by platelets when treated with 0.1 and 1 µM? (Figure 1B).

       Our reply: In the in vivo experiment we showed that 3 µM is the optimal concentration for the promotion of wound healing. For this reason, the cGMP was experiment was done with that concentration - we have clarified this in the text (see Materials and Methods - 2.4).

7.     Figure 1D. Why this chart does not include TOP-N53 treatment? The corresponding figure legend indicates that both, Figure 1C and 1D represent cell treatments with the different vehicles and formulations.

       Our reply: Results from treatments with TOP-N53 in DMSO and in the formulation were added to the figure.

  1. Since riociguat is the active substance, it does not be written with a capital letter.

       Our reply: This has been corrected.

  1. Figure 1 legend. This reviewer considers that "(C,D)" is unnecessary, because the figures "C" and "D" are indicated immediately after. Same for the other figure legends.

       Our reply: This has been changed as requested.

  1. In some figures, the charts did not include asterisks depicting statistical significance. If the compared groups do not exhibit any significant difference please avoid the text “*P≤ 0.05, **P ≤ 0.01, ***P ≤ 0.001, ****P ≤ 0.0001” in the figure legend.

       Our reply: This has been corrected.

  1. This reviewer considers that Supplementary figures S1 C and D are relevant for the findings, I suggest to incorporate to Figure 3 or rearrange Figure 3 and create another figure where charts be grouped.

       Our reply: The effect of TOP-N53 on inflammation was very minor and the effect on macrophages was not even significantly different. Therefore, we decided to show these "negative data" in the Supplement. We prefer not to include it in Fig. 3, because this figure is already very crowded.

12.  Figure 3. Re-epithelialization does not seem be lower in the TOP-N53 10µM formulation than the 1µM one. If you compare epidermal tongues (arrows) in the representative photomicrographs the difference is very clear.

       Our reply: This is correct, and we noticed that we had not included a representative wound. We now show another wound that reflects the average wound reepithelialization rate.

13.  Figure 3. To be more explicit, please include dashed lines and acronym indicators (WED, WB and WED) in TOP-N53 1 and 10µM treated tissues.

       Our reply: This has been done.

  1. Figure 5. Asterisks are too small to recognize them in the chart.

       Our reply: The asterisks were enlarged.

  1. This reviewer considers that Discussion section should include a brief analysis and comparison between DMSO and hydrogel formulations for TOP-N53.

       Our reply: We had already mentioned this in the discussion of the initial version, and we have now extended the discussion on this point.

16.  Figure S2. Please indicate surface and deep vessels above the LSOM images.

       Our reply: This has been done.

Round 2

Reviewer 1 Report

The manuscript is sufficiently improved in revision. The conclusion is clear and well supported by the data.